# The potential deployment of a pan-tuberculosis drug regimen in India: A modelling analysis

Nimalan Arinaminpathy[1]*, Gabriela B. Gomez[2], Kuldeep S. Sachdeva[3], Raghuram Rao[3], Malik Parmar[4], Sreenivas A. Nair[5], Kiran Rade[4], Sameer Kumta[6], David Hermann[7], Christy Hanson[7], Daniel P. Chin[7], Puneet Dewan[8]

**1** MRC Centre for Global Infectious Disease Analysis, Faculty of Medicine, Imperial College London, United Kingdom, **2** Department of Global Health and Development, London School of Hygiene and Tropical Medicine, London, United Kingdom, **3** Central TB Division, Government of India, New Delhi, India, **4** India Country Office, World Health Organization, New Delhi, India, **5** Stop TB Partnership, Geneva, Switzerland, **6** Bill and Melinda Gates Foundation, India Country Office, New Delhi, India, **7** Bill and Melinda Gates Foundation, Seattle, WA, United States of America, **8** Global Good, Intellectual Ventures, Seattle, WA, United States of America

* nim.pathy@imperial.ac.uk

**Data Availability Statement:** All relevant data are within the manuscript and its Supporting Information files.

## Abstract

There is increasing interest in future, highly-potent 'pan-TB' regimens against tuberculosis (TB), that may be equally effective in both drug-susceptible and rifampicin-resistant (RR) forms of TB. Taking the example of India, the country with the world's largest burden of TB, we show that adoption of these regimens could be: (i) epidemiologically impactful, and (ii) cost-saving to the national TB programme, even if the regimen itself is more costly than current TB treatment. Mathematical modelling suggests that deployment of a pan-TB regimen in 2022 would reduce the annual incidence of TB in 2030 by 23.9% [95% Bayesian credible intervals [CrI] 17.6–30.8%] if used to treat all TB cases, and by 2.30% [95% CrI 1.57–3.48%] if used to treat only RR-TB. Notably, with a regimen costing less than USD 359 (95% CrI 287–441), treating all diagnosed TB cases with the pan-TB regimen yielded greater cost-savings than treating just those diagnosed with RR-TB. One limitation of our approach is that it does not capture the risk of resistance to the new regimen. We discuss ways in which this risk could be mitigated using modern adherence support mechanisms, as well as drug sensitivity testing at the point of TB diagnosis, to prevent new resistant forms from becoming established. A combination of such approaches would be important for maximising the useful lifetime of any future regimen.

## Introduction

The need to initiate all TB patients on appropriate therapy, and to ensure relapse-free survival, is a key foundation of efforts to combat TB today [1,2]. Given this reliance on therapeutic strategies, drug resistance poses a critical issue, with an estimated 558,000 incident cases in 2017 being rifampicin-resistant (RR) [3]. Such forms of TB are both difficult to diagnose and to

**Funding:** NA received a grant from the Bill and Melinda Gates Foundation (BMGF), number OPP1095710. SK, DH, CH and DPC are current employees and PD was a former employee of BMGF. All contributed to model validation and to the final manuscript draft. The funder otherwise had no role in the study.

**Competing interests:** The authors have declared that no competing interests exist.

treat. Although RR-TB accounts for less than 6% of global TB burden, less than a quarter of that burden is ever found and treated, while the management of those RR-patients who are found and treated still accounts for over a quarter of programmatic spending on TB worldwide [3].

Second-line regimens show considerable toxicity with a success rate of only 50%, despite costing over a hundred times as much as first-line therapy, and lasting up to 24 months [1]. Moreover, of the estimated 330,000 cases of RR- or multi-drug-resistant TB amongst notified TB cases in 2017, only 49% were reported as having rifampicin resistance, the remainder presumably being initiated on inappropriate, first-line therapy. The advent of rapid molecular tests [4,5] have offered the opportunity for rapid recognition of drug sensitivity status, but uptake has been relatively slow, and limited by the available resources and capacity for both testing and subsequent treatment of detected RR-TB cases.

There is increasing interest in shorter, safer and more effective TB drug regimens to overcome these challenges [6]. Important developments in recent years include: new, shortened regimens for multi-drug-resistant TB [7]; recent, promising results from a new, 6-month regimen for extensively drug-resistant TB [8]; and several other new regimens now progressing through phase 3 clinical trials [9]. In future, anti-TB regimens could become sufficiently short, well-tolerated and effective that they could be considered even for management of drug-susceptible forms of TB. However, the optimum deployment of such a 'pan-TB' regimen may be determined as much by its cost as by its performance.

Some key arguments for and against pan-TB regimens have been presented elsewhere [10,11]. The development of a pan-TB regimen will undoubtedly face considerable clinical, technical and implementation challenges. Nonetheless, to assist strategic planning for the potential development of these regimens, there is a need to examine their potential epidemiological and cost implications in a given setting. Here, with a focus on India, we aim to address this need using a mathematical modelling approach. India is important for the future deployment of any new regimen: from an epidemiological perspective, improved efforts to combat TB in India will have implications for global TB control, as it accounts for approximately a quarter of global TB and RR-TB burden, respectively [3,12]; from a commercial perspective, no market for TB drugs would be complete without serving India.

In collaboration with the Revised National Tuberculosis Control Programme (RNTCP), we developed a mathematical model of TB transmission dynamics in India. The presumed characteristics of a future pan-TB regimen were informed by the WHO Target Regimen Profile [13]. Incorporating these into the transmission model, we sought: (i) to compare the health impact and incremental costs of a true pan-TB regimen (used for all TB) versus limiting use to those patients detected to have rifampicin resistance, (ii) to understand which regimen properties were the most impactful in the India setting, and hence potentially important to prioritize in development, and (iii) to determine the cost of a pan-TB regimen that could justify its adoption in budget terms. Additionally, we examined how the optimal deployment of a future regimen may be shaped by concurrent shifts in TB control in India, focusing on the example of private sector engagement.

## Methods

We developed a deterministic, compartmental model to capture the TB epidemic in India (see supporting information for technical details). The model distinguishes the public and private healthcare sectors, an important feature of the complex health system in India; it captures the acquisition and transmission of drug resistance, as well as the extent to which a TB patient's drug resistance status is identified in time to guide appropriate (first- or second-line) therapy.

As with any modelling framework, it is necessary to include simplifications: the model ignores age structure as well as the difference between different forms of TB (such as smear status and pulmonary vs extrapulmonary TB), essentially taking an average infectiousness over these forms. We also neglect HIV/TB coinfection, which accounts for an estimated 3% of TB burden in India [3].

We calibrated the model to: pooled subnational prevalence surveys from India [14,15]; estimates of the annual risk of infection from nationally representative infection surveys [16]; and WHO estimates for the proportion of cases that have rifampicin- or multi-drug resistance [3]. Each of these inputs has uncertainty attached. Using Bayesian melding [17], we incorporated this uncertainty into the model estimates, alongside uncertainty in other model inputs such as the quality of care in the private sector, and costs (see S2 Table in S1 Appendix). S1 Fig in S1 Appendix shows the resulting model fits to these calibration targets, along with uncertainty. We used the same Bayesian framework to quantify uncertainty in model projections, under the different intervention scenarios described in the main text.

Table 1 shows the series of key 'properties' that we chose to characterise a future pan-TB regimen. These properties required noteworthy assumptions: in particular, we assumed that a pan-TB regimen may–by virtue of its shortened duration–facilitate treatment completion. We assumed conservatively that a pan-TB regimen has the same hazard rate of loss-to-followup and mortality as for current, first-line therapy; its treatment completion rate is therefore promoted by its shorter duration, assumed here to be 2 months. We additionally assumed that more forgiving regimens could further improve efficacy. Recent meta-analysis of drug trial data [18] indicates that an independent risk factor for post-treatment recurrence risk is the quality of dosing implementation, i.e. missed doses, over the duration of treatment. Under routine programmatic conditions it is challenging to reach comparable levels of dosing implementation to those achieved under stringent clinical trial conditions. As a result, treatment effectiveness observed in programmatic settings has been less than efficacy observed in a clinical trial setting. We hypothesised that the efficacy-effectiveness gap could be mitigated by a future, short pan-TB regimen containing new drug molecules that are more 'forgiving' of missed doses than current, rifampicin-based regimens. If current levels of recurrence in India

**Table 1. Key regimen properties to be modelled, as relevant to the BMGF product development portfolio.** We modelled the impact of such an 'idealised' pan-TB regimen in India; we also modelled the impact of a 'less-than-ideal' future regimen, that meets all but one of these criteria.

| Pan-TB regimen property | Mechanism or rationale | Modelled effect |
|---|---|---|
| 1. Treatment initiation | Simplified regimens, without need for DST, minimise opportunities for initial loss to followup, between diagnosis and treatment initiation | Treatment initiation rates in public sector increased to assumed 95% [1] |
| 2. Treatment success for drug susceptible patients | Shorter, safer regimens enable more patients to successfully complete treatment without side effects | Assume 2 month regimen duration. Conservatively, assume same hazard rate of loss-to-followup as with current first-line therapy. Owing to shorter duration, treatment completion increases to 95% [1] |
| 3. Forgiveness of missed doses | Risk of relapse after treatment success increases substantially with poor adherence to medication intake [18]. New regimens with low risk could be more 'forgiving' of poor adherence. | Assume that recurrence rates for both drug-susceptible and RR-TB are halved relative to those on current first-line therapy, independent of treatment completion (i.e., reducing efficacy-effectiveness gap by half) [1,2] |
| 4. Treatment success in RR-TB patients | Owing to use of new molecules, future regimen would be equally effective in those sensitive and resistant to current first-line regimens | Treatment outcomes in RR-TB assumed to be equivalent to those in drug susceptible TB [1] |

[1] See S2 Table in S1 Appendix for baseline values for these outcomes under current regimens.

[2] We assume that treatment non-completion is associated with temporary bacteriological suppression, but an elevated recurrence risk, compared to those completing treatment [35,36].

are at least partly attributable to challenges in dosing implementation under programmatic conditions, we assumed that the efficacy-effectiveness gap could be halved by a future, more 'forgiving' regimen.

Further, we distinguished two deployment scenarios: (i) 'RR-only indication', meaning that the regimen is used only to replace current second-line regimens (i.e. for those recognised as having RR-TB under current coverage of DST), and (ii) 'universal indication', meaning that the regimen is used for all TB patients, regardless of drug susceptibility to rifampicin. We assumed that the regimen is deployed amongst all patients treated by the RNTCP (but not in the private sector), with coverage being scaled up over three years from an assumed introduction of the pan-TB regimen in 2022. As a baseline, we used a 'status quo' scenario, with current standards of care continuing indefinitely. Using Bayesian melding, we modelled uncertainty on all model projections using 95% credible intervals (CrI). Further technical details are provided in the Methods, and in the supporting information.

We did not aim to estimate the cost of the pan-TB regimen itself. There is active, ongoing work aiming to address the potential cost implications of new TB treatment regimens, for example as a result of reduced need for hospitalisation. Our approach complements these efforts, by examining thresholds on regimen cost, for a pan-TB regimen to be cost-neutral to the TB programme. While previous analysis has also addressed cost effectiveness, our approach is deliberately conservative from the programmatic perspective, with the aim of maximising the impact of new drugs within existing budget envelopes—a key concern for many national TB programmes in high-burden settings. Table 2 shows unit cost data from the Global

**Table 2. Unit costs used in the model.** All costs were subject to +/- 20% uncertainty, captured by a lognormal distribution. Sources: Global Health Cost Consortium [19], unless indicated otherwise.

| Cost item | Value (USD 2017) | Notes/source |
|---|---|---|
| **Diagnosis costs** | | |
| Cost of diagnosis per symptomatic, passive case finding in public system | USD 15 [12–18] | One microscopy and one X-ray, four outpatient visits [37] |
| Cost of diagnosis per symptomatic including DST with GeneXpert, passive case finding in public system | USD 40.3 [26.5–39.8] | GeneXpert and two outpatient visits [37] |
| **Treatment costs** | | |
| Cost per patient-month of first-line treatment | USD 32.2 [25.8–38.6] | Eight outpatient visits during intensive phase, four outpatient visits during continuation phase, plus monitoring tests during two visits. Also includes adherence support interventions (e.g. 99DOTS) at $2.8 per patient-month and DBT support at $8.6 per patient-month [21]. Drug costs based on GDF for standard regimen at $5.30 per patient-month [38] |
| Cost per patient-month of second-line treatment | USD 248.4 [198.7–298.0] | 144 outpatient visits during intensive phase, 15 days of hospitalisation, and 72 outpatient visits during continuation phase, plus monitoring tests [21]. Drug costs based on GDF for standard regimen at $78.7 per patient-month [38] |
| **Private sector incentives (Fig 4 only)** | | |
| Notification of a TB case diagnosed per Standards of TB Care | USD 4.3 [3.4–5.1] | Incentives to private sector providers [21], with assumed 15% management fee added |
| Incentive per month of first-line TB treatment completed | USD 4.3 [3.4–5.1] | |
| Incentive per course of first-line TB treatment completed | USD 8.6 [6.8–10.1] | |

Health Costing Consortium, an international initiative to collect systematic cost data relating to TB [19]. We estimated the incremental spending under both of the intervention scenarios described above (RR-only vs universal indication), under a range of assumptions for the per-regimen cost.

## Results

### Epidemiological impact of a pan-TB regimen

Fig 1 (upper panels) show the modelled incidence projections under the two deployment scenarios. Table 3 shows the numbers of averted under each of these scenarios, between 2022 and 2030. The pan-TB regimen yielded only modest influence on TB incidence, if limited to the RR-only indication, but if expanded to a universal indication, could reduce annual incidence

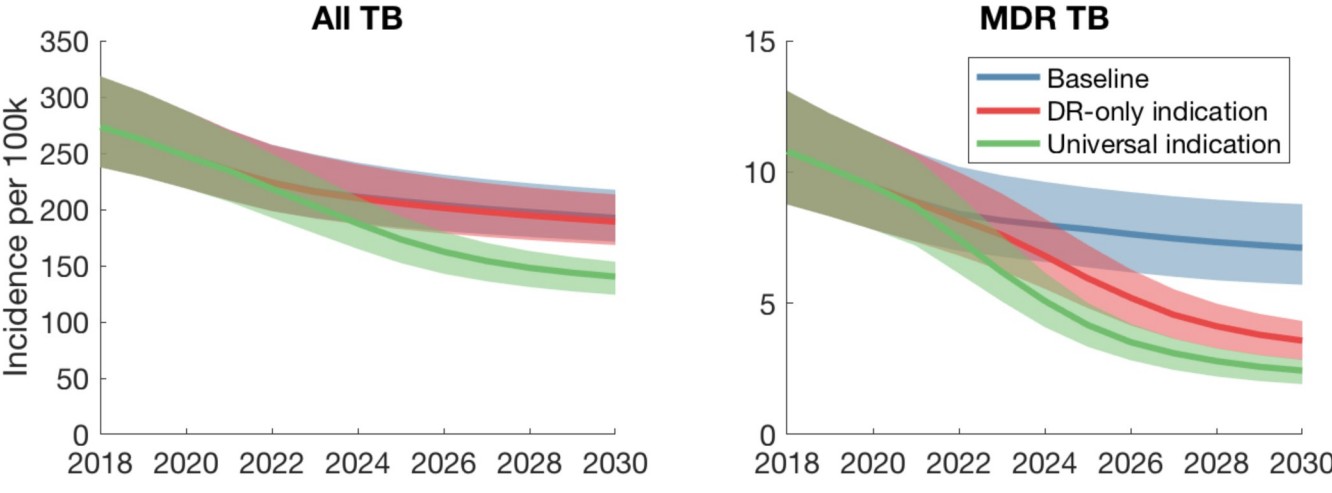

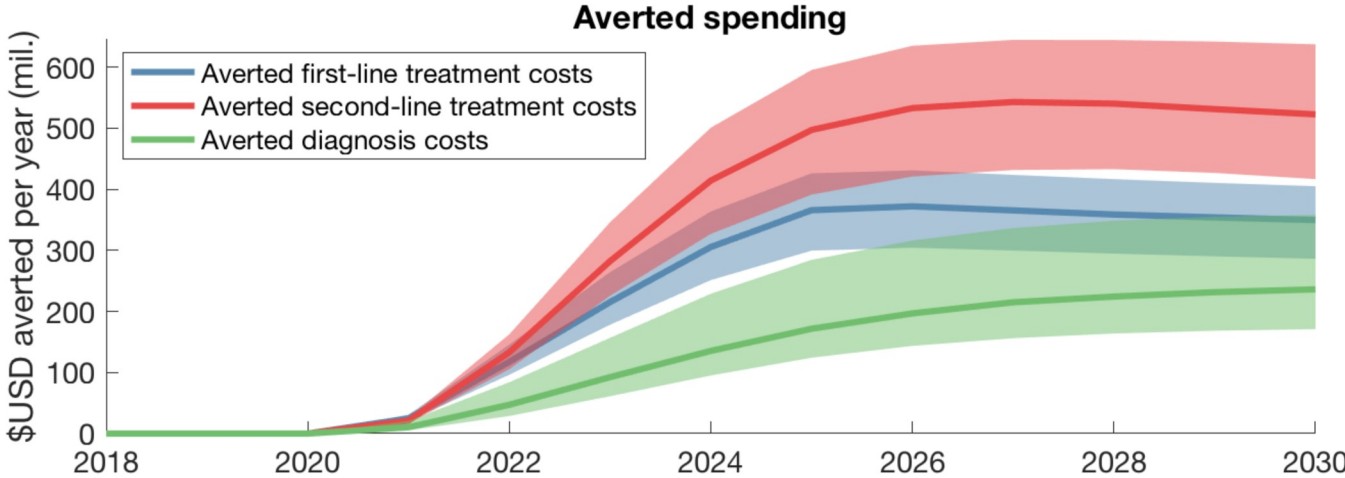

**Fig 1. Projected impact of a pan-TB regimen in India.** (A, B) Projections for all TB incidence (panel A) and RR-TB incidence (panel B), assuming that the regimen is introduced in 2022 and scaled up in a linear way over the subsequent three years. Table 3 shows the percent cases averted under the different deployment scenarios shown here. (C) Projected impact on 'conventional' programmatic spending (i.e. those not associated with the new regimen), under a pan-TB deployment scenario of universal indication. Diagnosis costs include diagnostic tests (smear microscopy as well as drug sensitivity testing with Xpert and culture in a certain proportion of cases). First- and second-line treatment include the costs of conventional drugs, as well as the staff costs involved in counselling and patient adherence support. Shaded areas show 95% credible intervals, while solid lines show median estimates. See Table 2 for details of cost inputs.

**Table 3. Summary of epidemiological and budget impact of a pan-TB regimen under the different scenarios modelled her.**

| Outcome | Baseline: status quo | | Baseline: private sector engagement | |
|---|---|---|---|---|
| | RR-only | Universal | RR-only | Universal |
| *Cases averted, millions* | 0.34 [0.233–0.502] | 4.07 [2.7–5.8] | 0.25 [0.18–0.35] | 4.11 [3.19–5.52] |
| *Non-pan-TB spending averted (first- and second-line drugs, and diagnosis), USD bn* | 4.19 [3.28–4.93] | 5.87 [4.77–7.03] | 3.97 [3.81–3.56] | 6.5 [6.42–5.9] |
| *Patient-months of pan-TB regimen used, millions* | 1.22 [0.888–1.62] | 19 [13.9–23.9] | 1.12 [0.879–1.35] | 24.8 [19.9–28.2] |
| *Threshold regimen cost for budget neutrality, USD* | 6386 [5849–6980] | 745.3 [636.2–872.2] | 7772 [7122–8200] | 762.2 [676.4–875] |
| *Threshold regimen cost for universal indication to dominate RR-only, USD* | 358.5 [286.7–441.4] | | 443.2 [374.1–529.1] | |

in 2030 by 23.9% [95% CrI 17.6–30.8%], compared to 'status quo' baseline in the same year. Similarly, while the RR-only indication reduced RR-TB incidence in 2030 by 54.2% [47.8–60.6], the universal indication could reduce RR-TB incidence in 2030 by 74.3% [95% CrI 68.1–80.3%]. Below we discuss potential reasons for this predicted behaviour.

## Threshold regimen costs for programmatic cost savings

We next examined the regimen cost threshold, in order for it to be cost-saving for the programme. The two deployment strategies (RR-only and universal indications) yield strikingly different thresholds. To illustrate this, we first examined the impact of a pan-TB regimen on conventional programmatic costs: that is, the averted spending on diagnosis, and costs of 'conventional' (first- and second- line) treatment costs. Fig 1 (lower panel) shows this averted spending, illustrating that the cost of conventional second-line drugs is a major component of the programmatic spending that could be averted by deployment of a pan-TB regimen.

Second, comparing these averted costs against the regimen-associated outlay, we estimated the incremental programmatic spending between 2022 and 2030, under a range of values for the per-regimen cost. Fig 2 shows resulting estimates under both deployment scenarios, with the dashed line indicating the level at which a pan-TB regimen will be cost-neutral to the programme. In particular, a RR-only indication would be cost-saving to the programme as long as the per-regimen cost is less than USD 6386 (95% CrI 5849–6980) (point *A*). A universal indication could also be cost saving (point *B*) as long as the regimen costs less than 745.3 (95% CrI 636.2–872.2). Notably (point *C*), if the regimen cost could be reduced still further to 358.5 (95% CrI 286.7–441.4), then a universal indication would be *more* cost-saving to the programme, than a RR-only indication.

## Identifying critical regimen characteristics

While Table 1 lists the properties of an idealised pan-TB regimen, in practice it is likely that future regimens would meet only some of these conditions and not others. To inform which properties are most critical for a future regimen, we performed analyses for two types of 'less-than ideal' (LTI) regimens, similar to that adopted in ref [20]: first, we modelled the impact (percent cases averted between 2022 and 2030) of an LTI regimen fulfilling *only one* of the properties in Table 1. Second, we modelled the impact of an LTI regimen fulfilling *all but one* of the properties in Table 1. By varying the property in question, we assessed the impact of each type of LTI regimen. In particular, if an idealised and LTI regimen avert, respectively, $X\%$ and $Y\%$ of cumulative cases between 2022 and 2030, we quantified the shortcoming of the LTI as simply $(X-Y)/X$. Results in Fig 3 illustrate that, for overall TB burden, the most important regimen property is that of forgiveness. For RR-TB impact, the most important regimen property is the cure rate that could be achieved amongst RR-TB patients. Below, we discuss the interpretations of these criteria, in relation to the challenges faced in control of drug-susceptible and RR-TB.

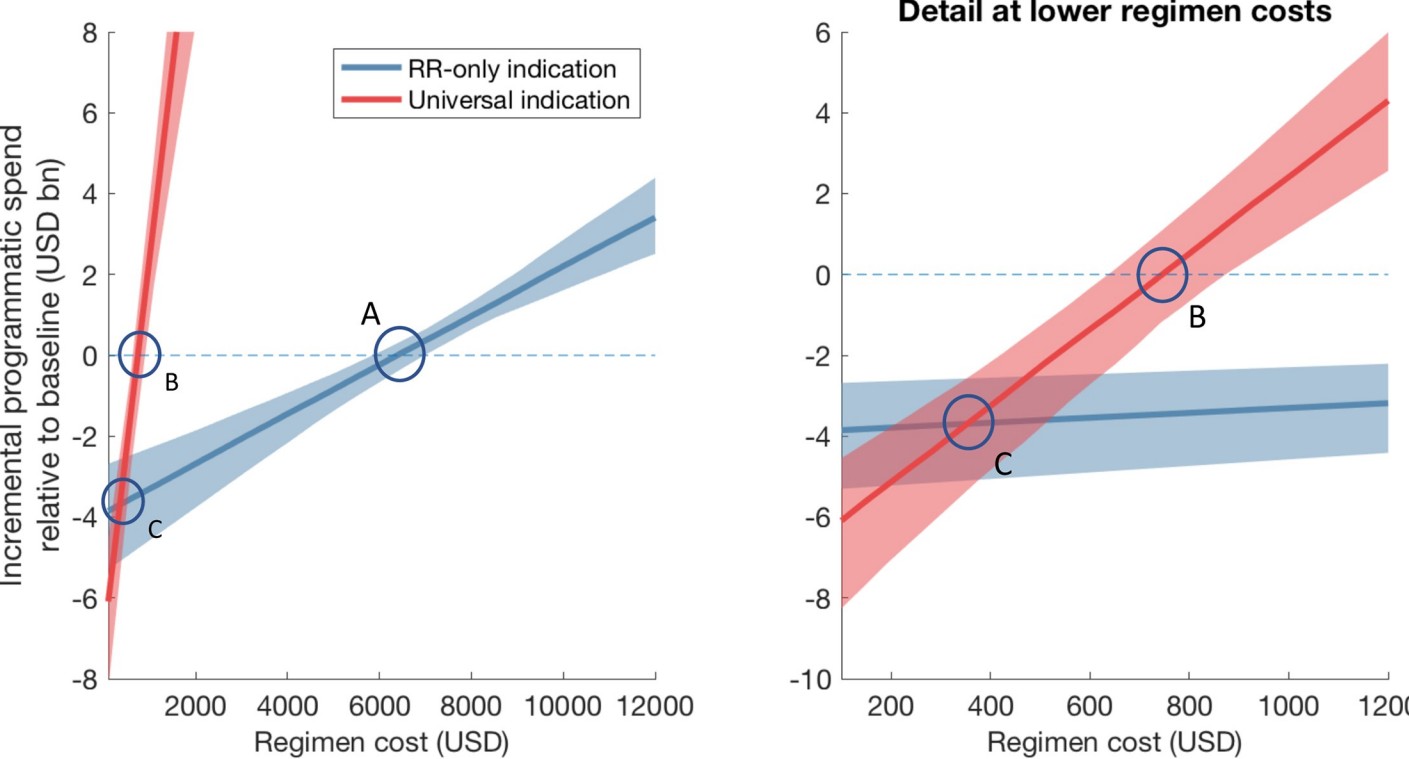

**Fig 2. Projected incremental cost to the India national TB programme between 2022 and 2030, from implementation of a pan-TB regimen, for different assumed values for the regimen cost (x-axis).** The right-hand panel shows a close-up of the left-hand, at lower regimen costs. The horizontal, dashed line shows the level at which a pan-TB regimen would be overall cost-neutral to the programme. Shaded areas show 95% credible intervals, while solid lines show median estimates. If a pan-TB regimen is used as a replacement for current second-line therapy, it will be cost-saving as long as it costs less than USD 6386 (95% CrI 5849–6980) (left-hand panel, point 'A'). If the regimen cost could be reduced to below USD 358.5 (95% CrI 286.7–441.4) per regimen, then it is *more* cost-saving to deploy with a universal indication, than with a RR-only indication (point 'B').

### Incorporating private sector engagement

With potential shifts in TB control over the coming decade [21], it is important to anticipate how these changes may affect our findings. We focused on a critical component in India's National Strategic Plan (NSP), where initiatives are currently underway to engage private providers into nationally coordinated TB control efforts [22–24]. Future regimens may therefore be used not only to treat patients managed by RNTCP, but also those being managed by engaged private providers. In our sensitivity analysis to this future scenario, we simulated a baseline in which 85% of providers are engaged between 2019–2022, with their standard of TB care (accuracy of diagnosis and treatment completion rates) being raised to the same standard as the public sector. Relative to this baseline, we assumed that the pan-TB regimen is deployed amongst all engaged private providers in addition to the public sector. Fig 4 shows resulting model projections. In the context of these background changes, a RR-only indication would be cost-saving with a regimen costing less than USD 7772 (95% CrI 7122–8200) per regimen; a universal indication would be *more* cost-saving with a regimen costing less than USD 443.2 (95% CrI 374.1–529.1) per regimen.

### Parameter sensitivity analysis

Concentrating on the threshold regimen costs illustrated in Fig 2, we performed a sensitivity analysis to different model assumptions. First, motivated by the leading importance of regimen

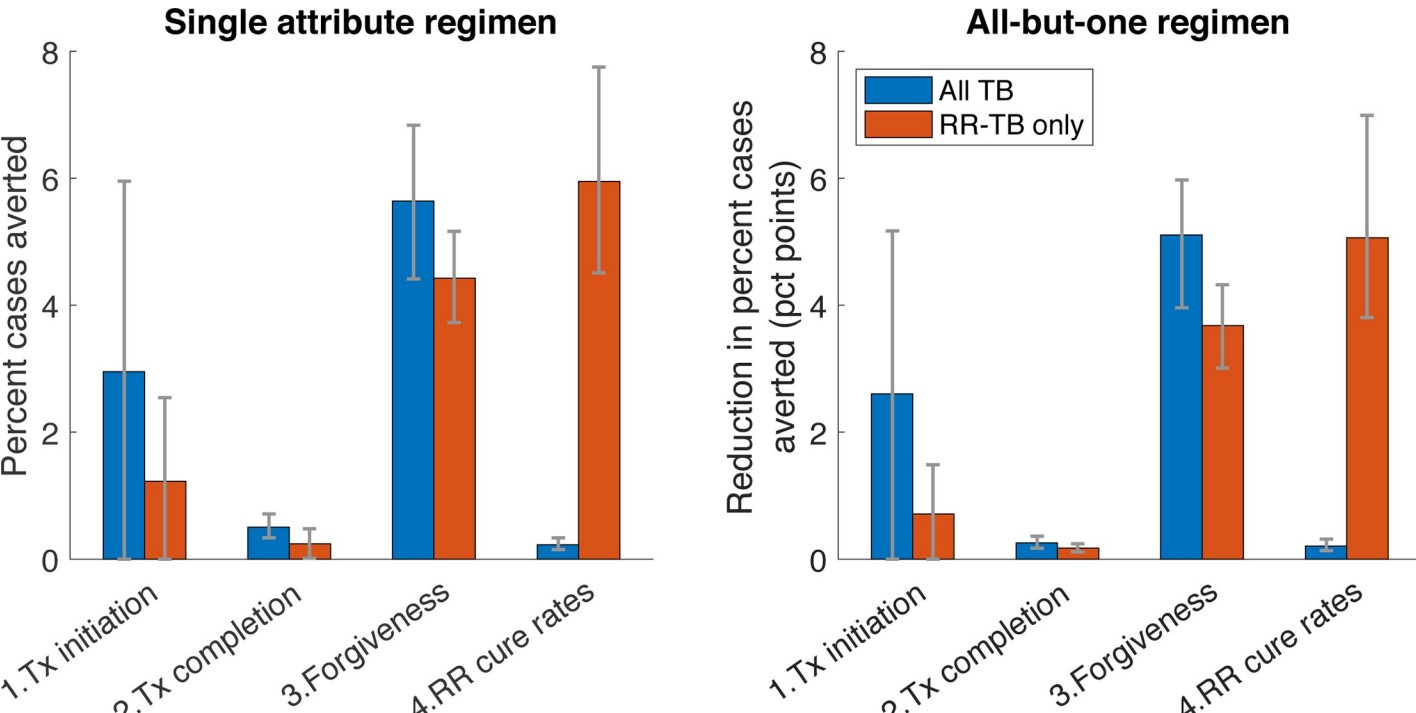

**Fig 3. Influence of different regimen properties.** Numbered 'properties' on the x-axis correspond to those listed in Table 1. As described in the main text, we first calculated the 'impact' X of an idealized regimen (percent cases averted between 2022 and 2030), one satisfying all the conditions listed in Table 1 (see Table 3 for impact estimates). We then simulated the impact Y of a 'less-than-ideal' (LTI) regimen: for any given regimen property of interest, we then quantified its 'attributable impact' as (X-Y)/X: that is, the proportional reduction in incidence impact, that results from a regimen failing to meet this criterion. Thus, higher bars represent properties that are more important for epidemiological impact. Red bars show an LTI regimen that meets only one of the regimen requirements in Table 1, while blue bars show an LTI regimen that meets all but one of the regimen requirements.

forgiveness in Fig 3, we estimated how the threshold regimen cost would change under a regimen that meets all of the attributes listed in Table 1, with the exception of forgiveness. We repeated this analysis with treatment initiation and RR-TB cure rates (also of leading importance for all-TB impact and RR-TB impact, respectively), as well as with the regimen duration (assumed in the main analysis to be 2 months, and in this sensitivity analysis to be 4 months). Results, shown in Fig 5, illustrate that the threshold regimen cost is most sensitive to the RR-TB cure rate, under a universal indication, and relatively insensitive to other regimen characteristics. Below we discuss the interpretation of these findings.

Finally, all uncertainty in model projections arises from the parameter and input uncertainty summarised in S2 Table in S1 Appendix. We aimed to identify which model inputs are most influential in this overall uncertainty. S2 Fig in S1 Appendix shows the partial rank correlation of a selected model output, against model inputs (data and parameters). Inputs associated with the highest correlation coefficients are the most 'important' for model projections, in the sense that improved precision in these inputs would yield the greatest improvement in the precision of model outputs. The figure illustrates that the leading contributors to uncertainty in model projections are related to the epidemiological data used as calibration targets. Amongst those parameters specified by assumptions in the absence of systematic data (principally those relating to the private sector), the most important is the rate of initial loss to follow-up amongst private providers.

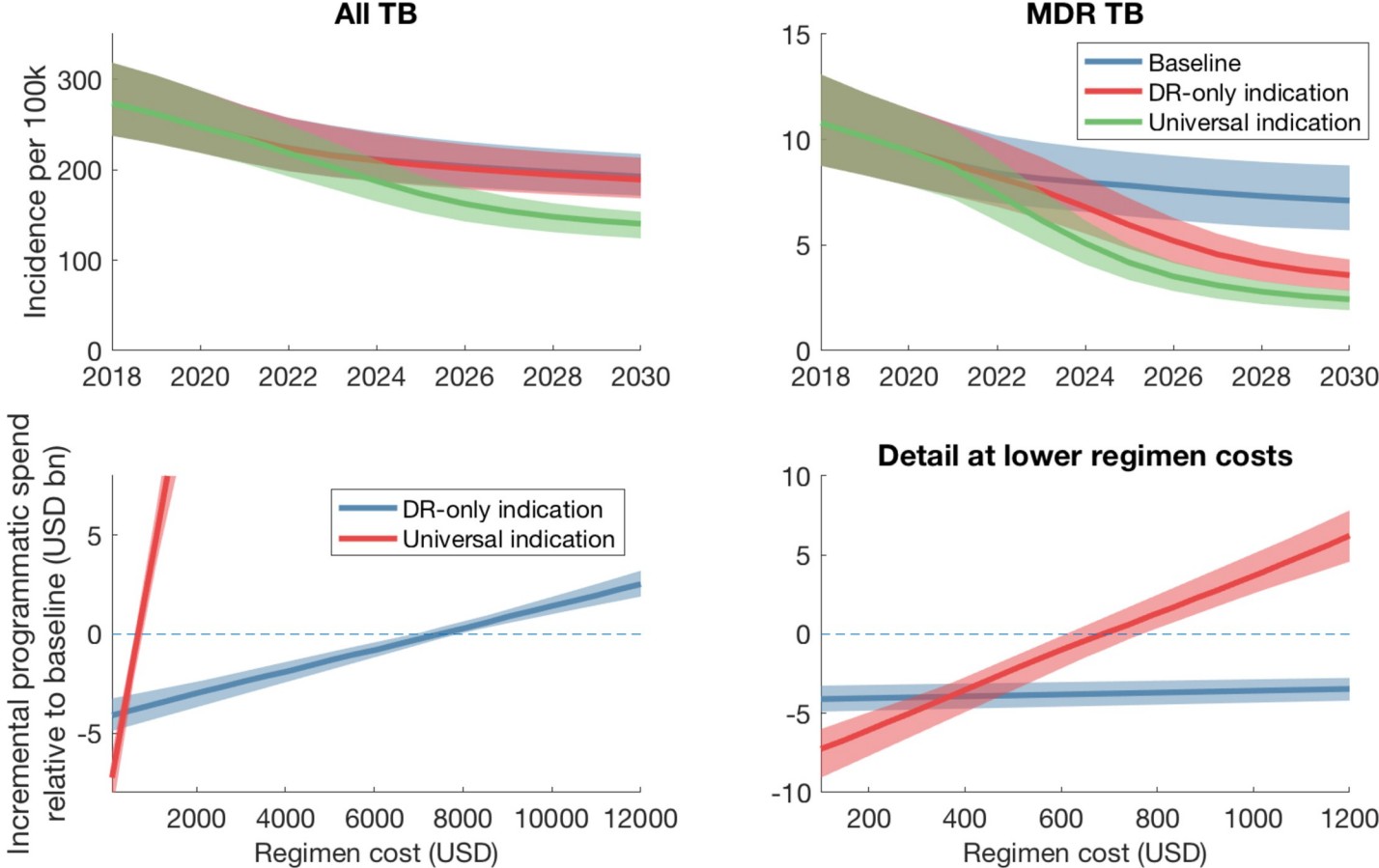

**Fig 4. Model projections under an alternative baseline (private sector engagement), of initiatives to engage with 85% of private healthcare providers in India between 2019 and 2022.** We assume that engaged private providers have their quality of TB care (accuracy of diagnosis, levels of DST and treatment success) raised to the same standard as in the public sector (see also S2 Table in S1 Appendix); and further that the pan-TB regimen will be deployed amongst all engaged private providers in addition to the public sector. Upper panels show epidemiological impact, as in (Fig 1A and 1B). Lower panels show incremental programmatic spending for a range of scenarios for regimen cost, as in Fig 2.

## Discussion

Planning for the design, development and deployment of future regimens would benefit greatly from estimates for their population-level impact, both on efforts to combat TB and on country TB budgets. Here, using mathematical models of TB transmission, we have aimed to address these impacts in the specific context of India: a setting that has key importance for global efforts to end TB, as well as for the potential market for future regimens.

Overall, our results suggest that deployment of a future pan-TB regimen could have a meaningful impact on TB incidence in India (Fig 1). Notably, although a RR-only indication reduces annual RR incidence by 54% by 2030, a universal deployment reduces RR incidence by a further 20 percentage points in the same time period. This is because, unlike a RR-only indication, deploying a pan-TB regimen universally would benefit the substantial numbers of RR-TB cases who would otherwise have been initiated on inappropriate, first-line therapy.

Regimen costs are a complex interplay between costs and compassion, production and politics, markets and marketing; in the present study, we deal with none of those. Instead we ask, 'at what cost threshold does the fiscal argument become overwhelmingly supportive of change?' We find that a pan-TB regimen, if used as a replacement for current second-line

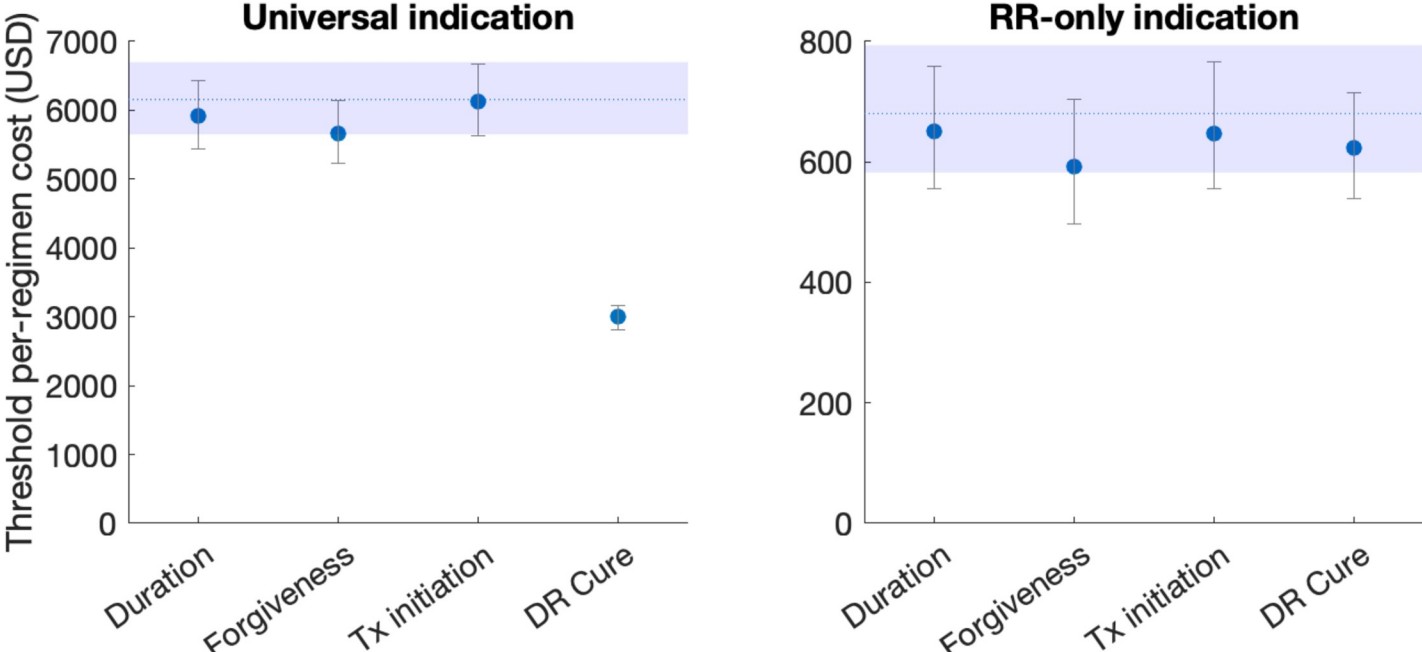

**Fig 5. Sensitivity analysis of threshold regimen cost, in order for a pan-TB regimen to be cost-saving to the programme, in relation to different assumptions for regimen characteristics.** We model 'all-but-one' regimen scenarios, as in Fig 3, but here focusing on the most prominent attributes in Fig 3, along with regimen duration. While the latter is assumed to be 2 months in the main analysis, here we consider a regimen of 4 months duration. Results suggest that—although threshold regimen costs appear relatively robust to these different characteristics, the one exception is RR-TB cure rate under a universal indication. For this attribute, a decrease to 50% (from 85%) is sufficient to bring the threshold cost down by roughly half (that is, posing a more stringent condition for budget neutrality). These results highlight that the economic value of a pan-TB regimen would derive largely from its ability to benefit patients with RR-TB, who might otherwise receive inappropriate therapy.

therapy, could be cost-saving to RNTCP if it costs less than USD 6386 (95% CrI 5849–6980) per patient treated. This total regimen cost is comparable to current second-line regimens (at roughly USD 5960), although the per-month cost is over 10 times as much as current regimens. However, the fiscal case for universal deployment will be particularly strong if the regimen cost could be reduced to below USD 358.5 (95% CrI 286.7–441.4) (compared to approx. USD 190 for current first-line regimens), at which point it would be more economical for the programme to treat all TB cases with the new regimen, rather than just those with rifampicin resistance. These estimates changed only modestly when assuming an alternative baseline, of comprehensive engagement with the private sector. Overall it is particularly notable that a more-expensive regimen could overall be cost-saving, especially if the envisioned improvements in efficacy were to be realised under programmatic usage. Critically, per-regimen cost is not the only determinant of incremental programmatic spending: a pan-TB regimen could facilitate cost savings in other important yet indirect ways, principally the averted spending that arises from the regimen's epidemiological impact on TB and RR-TB (Fig 1C).

Our work builds on earlier modelling analysis of the potential impact of pan-TB regimens [10,25], particularly one study that also addressed the economic case for the introduction of such regimens [25]. Our findings, for the threshold cost of a new regimen that achieve overall cost savings, are broadly consistent with these results, and emphasize the consistent prediction of important health impact of a universally-deployed pan-TB regimen. However, our work builds on these foundations in three key ways. First, our results also illustrate a new, programmatically important insight: that despite a higher initial investment, it is possible for a high-coverage, universal indication to be more cost-saving to the TB program budget than a RR-only indication (point C in Fig 2), even in a brief (8-year) time horizon. Second, while the

earlier study [25] analysed a generic TB epidemic, the present work is specifically focused on India, demonstrating that the general predictions apply even where the health system context could reasonably have undermined the value of a regimen change. In particular, we have incorporated the private healthcare sector, a critical feature in the Indian context, and addressed cost thresholds for a new regimen depending on its deployment among patients treated in this sector (Fig 4). Third, the most important performance characteristics of a future regimen are likely to depend on the setting in which it is deployed. With a focus on the India setting, we have identified which regimen characteristics could matter most for impact (Fig 3), which may prove useful when the inevitable trade-off and choices emerge in pan-TB regimen optimization.

In terms of reductions in TB incidence under a universal indication, we identified 'forgiveness' to missed doses as the most influential regimen attribute for impact on overall TB burden, and treatment outcomes in RR-TB patients as most influential for impact on RR-TB burden (Fig 3). Notably, when focusing on the threshold cost for budget neutrality, the only influential attribute is treatment outcome in RR-TB patients, under a universal indication (Fig 5). As discussed above, many patients with RR-TB are placed on inappropriate, first-line therapy. It is only with a universal indication that these patients would benefit from a pan-TB regimen: the results in Fig 5 thus illustrate that the economic value of a future pan-TB regimen will derive largely from its ability to cure these individuals.

As described above, these properties may well vary by setting: for example, on the burden of drug resistance, and on existing levels of drug susceptibility testing. Future work should address how critical regimen characteristics may vary in other high-burden settings, for example in countries of the former soviet union, where drug resistance accounts for a substantially higher proportion of overall TB burden [3].

As with any modelling study, our work has some limitations to note. On the costing side, although cost-effectiveness is an important consideration for national TB programmes, we have focused here on the more stringent condition of cost saving, to the TB programme. As noted above, our approach is deliberately conservative from the programmatic perspective: nonetheless, future analysis could also incorporate other perspectives including societal costs, which we have ignored here. Such costs are likely to favour the cost-effectiveness of a pan-TB regimen, for example by taking account of the lost income and household expenditure arising from inappropriate first-line treatment, and from long-duration, second-line treatment: these are examples of costs that could be averted by a new regimen. Second, we do not consider the costs of implementing a new regimen throughout the health system, which would offset the costs averted shown in Fig 1C. Further work could address the extent to which these implementation costs modify the threshold regimen costs shown in Fig 2. On the transmission modelling side, our model necessarily entails several simplifications, as listed above in the model description.

The value of a "pan-TB" regimen has been put into question because of the likelihood that resistance to any new drugs would eventually develop [11]. A limitation of our present analysis is that it does not address these issues of drug resistance. On the one hand, although resistance to isoniazid or rifampin began to emerge a few years after their introduction as part of standardized treatment regimens [26,27], this occurred at a time when little was done to prevent the emergence of drug-resistance. Today, national TB control programs are using different approaches to slow the emergence of drug-resistance such as fixed-dose combination drugs, adherence technologies, and more intensive support of patients [28–30]. Furthermore, better-tolerated drugs could reduce treatment discontinuation and the risk of acquired drug resistance. To the extent that these and other measures can indeed lower the risk of drug resistance emerging, they would extend the value of a "pan-TB" regimen for several years.

Nonetheless, for any new regimen, it remains critically important to mitigate impact of the eventual emergence of resistance. With current anti-TB regimens, the use of drug susceptibility testing at the time of TB diagnosis has had an important effect in saving lives [31], and could be as important as the interventions discussed above, in slowing the spread of drug resistance [32]. Similar measures for future regimens may additionally prevent new, resistant forms from being established through transmission. Thus, the use of a pan-TB regimen does not mean that the current DST system for TB drugs should be dismantled. Ideally, such DST infrastructure would not be separate from that used for primary TB diagnosis. For example, molecular diagnostic tools are increasingly being used for TB diagnosis, while also offering the ability for test for genotypic signatures of drug resistance [33,34]. The increased adoption of these tools, while improving the accuracy of TB diagnosis, would concomitantly help to maintain the infrastructure needed, for improved drug sensitivity testing in future. In the present work, we note that our costing scenarios do not assume a discontinuation of DST; moreover, our scenario of private sector engagement (Fig 4) goes some way towards capturing a scenario of increased uptake of molecular diagnostic tools.

Another important consideration, that we have not addressed in the current work, is the potential for adverse events to a future drug regimen [11]. Particularly in the context of drug resistance, discussed above, some patients will suffer regimen toxicity without having clinical benefit from the regimen. Future work to address these factors would benefit from ongoing studies of regimens currently under development, to inform the levels of adverse events that might be expected of future drug regimens.

In conclusion, despite the considerable scientific, clinical and operational challenges involved, the successful development of a pan-TB regimen could have profound implications for TB control. A more effective regimen deployed to all TB patients, precluding DST altogether, could yield accelerated reductions in TB incidence, while being cost-saving overall for national programs within 10 years. In strategic planning for such regimens, it is important to take into account the epidemiological and health system context in which they will be deployed. Dynamical models, such as that presented here, could offer helpful approaches for doing so.

## Supporting information

**S1 Appendix.**
(DOCX)

## Author Contributions

**Conceptualization:** Nimalan Arinaminpathy, Kuldeep S. Sachdeva, Raghuram Rao, Sameer Kumta.

**Formal analysis:** Nimalan Arinaminpathy, Gabriela B. Gomez, Malik Parmar.

**Investigation:** Nimalan Arinaminpathy, Gabriela B. Gomez, Malik Parmar, Sreenivas A. Nair, Kiran Rade, David Hermann.

**Methodology:** Nimalan Arinaminpathy, Gabriela B. Gomez, Malik Parmar, Kiran Rade, Sameer Kumta, David Hermann, Christy Hanson.

**Resources:** Daniel P. Chin, Puneet Dewan.

**Supervision:** Daniel P. Chin, Puneet Dewan.

**Validation:** Kuldeep S. Sachdeva, Raghuram Rao, Malik Parmar, Sreenivas A. Nair, Kiran Rade, Sameer Kumta, David Hermann, Christy Hanson, Daniel P. Chin, Puneet Dewan.

**Writing – original draft:** Nimalan Arinaminpathy, Gabriela B. Gomez.

**Writing – review & editing:** Nimalan Arinaminpathy, Gabriela B. Gomez, Kuldeep S. Sachdeva, Raghuram Rao, Malik Parmar, Sreenivas A. Nair, Kiran Rade, Sameer Kumta, David Hermann, Christy Hanson, Daniel P. Chin, Puneet Dewan.

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
