## [Decision Letter · Decision Letter 0]

24 Oct 2019

PONE-D-19-22322

The potential deployment of a pan-tuberculosis drug regimen in India: a modelling analysis

PLOS ONE

Dear Dr Arinaminpathy,

Thank you for submitting your manuscript to PLOS ONE. After careful consideration, we feel that it has merit but does not fully meet PLOS ONE’s publication criteria as it currently stands. Therefore, we invite you to submit a revised version of the manuscript that addresses the points raised during the review process.

ACADEMIC EDITOR: Please fully address the comments raised by reviewer 1, taking into consideration the considerable concerns raised by reviewer 2.

We would appreciate receiving your revised manuscript by Dec 08 2019 11:59PM. To enhance the reproducibility of your results, we recommend that if applicable you deposit your laboratory protocols in protocols.io, where a protocol can be assigned its own identifier (DOI) such that it can be cited independently in the future. For instructions see: http://journals.plos.org/plosone/s/submission-guidelines#loc-laboratory-protocols

We look forward to receiving your revised manuscript.

Kind regards,

Helen Cox

Academic Editor

PLOS ONE

Journal Requirements:

2.  Please declare in the competing interests that one of the authors is an independent consultant.

Within your Competing Interests Statement, please confirm that this commercial affiliation does not alter your adherence to all PLOS ONE policies on sharing data and materials by including the following statement: "This does not alter our adherence to PLOS ONE policies on sharing data and materials.” (as detailed online in our guide for authors http://www.PLOSone.org/static/editorial.action#competing).

If this adherence statement is not accurate and there are restrictions on sharing of data and/or materials, please state these. Please note that we cannot proceed with consideration of your article until this information has been declared.

Additional Editor Comments (if provided):

Reviewers' comments:

Reviewer's Responses to Questions

**Comments to the Author**

1. Is the manuscript technically sound, and do the data support the conclusions?

Reviewer #1: Yes

Reviewer #2: No

2. Has the statistical analysis been performed appropriately and rigorously? 

Reviewer #1: Yes

Reviewer #2: No

3. Have the authors made all data underlying the findings in their manuscript fully available?

Reviewer #1: Yes

Reviewer #2: Yes

4. Is the manuscript presented in an intelligible fashion and written in standard English?

Reviewer #1: Yes

Reviewer #2: Yes

5. Review Comments to the Author

Reviewer #1: In this manuscript, the investigators use a mathematical model of TB natural history and transmission to ask several fundamental questions about how a novel treatment regimen for TB, that could be used to treat both pan-susceptible and RR-TB, could impact the TB epidemic in India over a short time horizon (<10 years). These questions include:

1) What is the projected impact of the regimen on TB and MDR-TB compared with the status quo?

2) How does the intended use of the new regimen (for RR-only or for all TB cases), affect such projections?

3) Which performance characteristics of the regimen [ie treatment initiation rates, treatment completion/success rates, reducing recurrent rates (as described by the authors “treatment forgiveness” of missed doses), and treatment success in RR-cases] are the most important for achieving epidemiological impact?

4) What is the cost threshold at which these new regimens would be cost neutral compared with status quo?

Specific comments:

1) This manuscript addresses a question being actively discussed by the donor community and thus serves an important purpose in advancing these conversations on the potential role of new regimens that can potentially be used without DST.

2) I commend the authors on an excellently written manuscript which is a pleasure to read. The major assumptions are clearly explained and limitations are not buried in supplemental material.

3) The modeling and economic analysis are expertly described – the simplicity of the model (in most cases) seems a reasonable match for the state of the science related to this question of pan-tuberculosis regimens which is, at this time, not an intervention ready for deployment.

4) The results presented in the main text present a very favorable case for the use of pan-tuberculosis regimens and deployment to all TB cases. In my view, this reflects a rather optimistic view on both the manner in which the new tuberculosis regimen may be adopted (reflected in the treatment initiation parameter) as well as the rates of treatment completion and the actual performance characteristics of the regimen (forgiveness to missed doses, efficacy for RR-TB). I recognize that there is a complex matrix of questions that can be asked, and I don’t expect that the authors would likely choose to address all of these, but several of the following concerns strike me as important to present a more balanced case for both funders and drug developers to consider.

a. While the authors conduct additional analysis in which they sequentially eliminate one of these 4 regimen properties to estimate the impact of a regimen that fails to meet a single one of these properties, it would seem of interest to ask the reverse: What would be the impact of regimens that meet only one of these properties?

b. The assumption of a 2 month regimen seems deserving of a sensitivity analysis – what if the regimen only could be shortened to 4 months for example?

c. The assumptions around the forgiveness of the regimen to missed doses strike me as a particular example of optimistic thinking and I would like to understand if there is more evidence in the literature to suggest that new regimens are likely to this property (and if so, how dramatic it will be). In my view, this is of critical importance especially since the authors found that this property was the most important for projections of reduction in overall TB burden. The fact that improving outcomes for all cases might be the most important feature of a new regimen does not surprise me and seems in line with other modeling of new regimens (eg Kendall et al PLoS Med 2017), but the idea that a new shorter regimen will be substantially better (2x) in reducing recurrence rate seems dramatic.

d. The lack of attention to the potential for emergent resistance to the new regimen is a real limitation. I realize this would seriously compromise the simplicity of the work, but stands out as a feature of concern given that one of the most serious concerns about DST-free regimens is this potential (as described here: Acquah and Furin Lancet Infect Dis 2019 and Dheda et al Lancet Resp Med 2018.). The authors do mention this as a limitation, but I think some additional text linking to some of the dissenting literature might help balance out the case and raise readers attention to the active controversy in this area.

Reviewer #2: This paper is on a potentially important topic, but I am having a hard time getting past the authors' definition of an "ideal regimen". What is described here is almost a fantasy regimen, and I do not understand how they chose the parameters (other than what is in the WHO target product profile--which is a bit of a "dream" document as well). We are nowhere near having a two-month regimen for any forms of TB, and with the exception of the TRUNCATE TB trial, I am not aware of any studies that are even attempting to look at this. As for a "forgiving regimen" that would allow people to miss doses with no implications for their outcomes or for the generation of resistance, this is a wonderful notion, but nowhere on the horizon that I can see. As for assuming that 95% will initiate therapy, there is no evidence that a regimen change alone will lead to increased case detection and initiation of therapy. And the authors also seem to assume this ideal regimen will not be associated with any adverse events and that there will be no pre-existing resistance to the medications in the regimen (which is not true for any drug--there are almost always naturally occurring mutations that are selected for with inadequate therapy). I am not sure such a regimen is even possible, yet these assumptions form the underpinnings of the entire modeling exercise. Why did the author select parameters that are so far outside the realm of what is possible to accomplish in the next decade in TB? If the paper wants to dream the impossible, why not a two week regimen? Some detailed justification would have to be given for why the authors selected the features of the "ideal regimen" that they did--beyond that this is part of a TB target product profile. Otherwise, it could appear that they selected regimen parameters that would give them the results they wanted. I realize there is room for adjusting parameters in modelling work--and in fact changing these parameters can be important in understanding the impact of certain interventions. But what the authors have done here really seems to strain the bounds of what is possible even in the most ambitious and optimistic views of the TB treatment landscape.

Second, the authors do not take into account the down sides of dismantling systems for carrying out drug susceptibility testing. By their own admission, this "universal regimen" would likely only work for several years before there was significant resistance. But with the goal of "obviating the need for DST", what would happen then? Would we not be back in the same situation we are when we gave the universal regimen of INH, RIF, PZA, and EMB and did away with DST? What would be the costs of needing to rebuild a system wherein DST could then be done to the drugs in the "ideal regimen". Exploring this may not be possible in the model but should surely be raised in the discussion section.

Third, the authors leave out any consideration of adverse events and the costs associated with those. In the absence of DST, there will be people who receive medications that their M. tuberculosis strain is resistant to. This is inevitable. And these individuals receiving nothing from those medications but toxicity. Perhaps the "ideal regimen" that the authors want to use also has no associated adverse events? But this is not ever going to be reality. So what about the costs of these unnecessary adverse events? Even if they cannot build them into the model, this issues should surely be mentioned in the discussion section.

Fourth, I am also puzzled as to why they would not account for the selection and amplification of drug resistance. This is a major problem with the use of "universal regimens" and must be accounted for in the model. The development and propagation of rifampicin resistance was a major problem with the use of the universal INH, RIF, PZA, and EMB regimen deployed without DST. Developing a model that pretends this is not relevant or not a risk perpetuates a fallacy and weakens the arguments in favor of a universal regimen.

There are also multiple smaller issues with this paper. First, the use of the term defaulter is stigmatizing and should be avoided. Second, the authors are citing literature incorrectly. For example, they cite the Schnippel et al paper from South Africa that showed a decreased mortality when bedaquiline was given to people living with RR-TB. Yet they claim this paper shows a "reduced mortality with shorter regimens." If anything, the data published on shorter regimens (most notably the Nunn et al trial published in the New England Journal of Medicine) showed a trend toward higher mortality among people treated with a shorter regimen, especially if they were also living with HIV.

I think a modeling paper on this topic could be very important and interesting. But it has to have some basis in reality and it cannot exclude "inconvenient" parameters that would most certainly play a role in the effectiveness of such a regimen (including side effects and resistance selection).

6. PLOS authors have the option to publish the peer review history of their article (what does this mean?). If published, this will include your full peer review and any attached files.

Reviewer #1: No

Reviewer #2: Yes: Jennifer Furin

---

## [Author Response · Author response to Decision Letter 0]

5 Jan 2020

Please see the file attached to this submission, 'Reviewer_responses.docx'

---

## [Decision Letter · Decision Letter 1]

20 Jan 2020

PONE-D-19-22322R1

The potential deployment of a pan-tuberculosis drug regimen in India: a modelling analysis

PLOS ONE

Dear Dr Arinaminpathy,

Thank you for submitting your manuscript to PLOS ONE. After careful consideration, we feel that it has merit but does not fully meet PLOS ONE’s publication criteria as it currently stands. Therefore, we invite you to submit a revised version of the manuscript that addresses the points raised during the review process.

ACADEMIC EDITOR: 

I see that the abstract has not been revised to reflect the more balanced view of the paper. Please add some text around the assumptions and limitations of the study.

There is a typo in line 289 'forimpact'.

Lines 328-329: I think it might be good to temper the statement the a pan-TB regimen will likely to be useful for several years. There is limited evidence that the introduction of fixed dose combinations and adherence technologies have reduced the risk of resistance acquisition.

We would appreciate receiving your revised manuscript by Mar 05 2020 11:59PM. To enhance the reproducibility of your results, we recommend that if applicable you deposit your laboratory protocols in protocols.io, where a protocol can be assigned its own identifier (DOI) such that it can be cited independently in the future. For instructions see: http://journals.plos.org/plosone/s/submission-guidelines#loc-laboratory-protocols

We look forward to receiving your revised manuscript.

Kind regards,

Helen Suzanne Cox

Academic Editor

PLOS ONE

Reviewers' comments:

Reviewer's Responses to Questions

**Comments to the Author**

1. If the authors have adequately addressed your comments raised in a previous round of review and you feel that this manuscript is now acceptable for publication, you may indicate that here to bypass the “Comments to the Author” section, enter your conflict of interest statement in the “Confidential to Editor” section, and submit your "Accept" recommendation.

Reviewer #2: (No Response)

2. Is the manuscript technically sound, and do the data support the conclusions?

Reviewer #2: Yes

3. Has the statistical analysis been performed appropriately and rigorously? 

Reviewer #2: I Don't Know

4. Have the authors made all data underlying the findings in their manuscript fully available?

Reviewer #2: Yes

5. Is the manuscript presented in an intelligible fashion and written in standard English?

Reviewer #2: Yes

6. Review Comments to the Author

Reviewer #2: Thank you for updating your paper to address all the comments and concerns. It is a stronger paper now and I think more based in reality.

I have only one small comment. In the paper there appears to be a missing reference--I think the authors meant to put one in here but it just says "ref".

Today, national TB control programs are using different

325 approaches to slow the emergence of drug-resistance such as fixed-dose combination drugs,

326 adherence technologies, and more intensive support of patients (ref). Furthermore, better

327 tolerated drugs could reduce treatment discontinuation and the risk of acquired drug

328 resistance. Therefore, the lower risk of drug resistance means that a “pan-TB” regimen will

329 likely be useful for several years.

While I agree with the overall message here, the authors should still note that one important strategy that has been used to slow the emergence of drug resistance has drug-susceptibility testing at the time of TB diagnosis. This is effective and is standard of care in wealthy countries. Non-adherence is not the only cause of emerging drug resistance but so is inadequate therapy. The authors need to acknowledge this.

7. PLOS authors have the option to publish the peer review history of their article (what does this mean?). If published, this will include your full peer review and any attached files.

Reviewer #2: No

---

## [Author Response · Author response to Decision Letter 1]

5 Mar 2020

Please see attached file, 'Response to reviewers'

---

## [Editor Report · Decision Letter 2]

10 Mar 2020

The potential deployment of a pan-tuberculosis drug regimen in India: a modelling analysis

PONE-D-19-22322R2

Dear Dr. Arinaminpathy,

We are pleased to inform you that your manuscript has been judged scientifically suitable for publication and will be formally accepted for publication once it complies with all outstanding technical requirements.

With kind regards,

Helen Suzanne Cox

Academic Editor

PLOS ONE
---

## [Editor Report · Acceptance letter]

16 Mar 2020

PONE-D-19-22322R2 

The potential deployment of a pan-tuberculosis drug regimen in India: a modelling analysis 

Dear Dr. Arinaminpathy:

I am pleased to inform you that your manuscript has been deemed suitable for publication in PLOS ONE. Congratulations! Your manuscript is now with our production department. 

With kind regards,

on behalf of

Prof Helen Suzanne Cox 

Academic Editor

PLOS ONE